# Assessing the drivers of syphilis among men who have sex with men in Switzerland reveals a key impact of screening frequency: A modelling study

Suraj Balakrishna[1,2]☯*, Luisa Salazar-Vizcaya[3]☯, Axel J. Schmidt[4,5], Viacheslav Kachalov[1,2], Katharina Kusejko[1,2], Maria Christine Thurnheer[3], Jan A. Roth[6,7,8], Dunja Nicca[9], Matthias Cavassini[10], Manuel Battegay[6], Patrick Schmid[4], Enos Bernasconi[11], Huldrych F. Günthard[1,2], Andri Rauch[3‡], Roger D. Kouyos[1,2‡], the Swiss HIV Cohort Study (SHCS)¶

1 Division of Infectious Diseases and Hospital Epidemiology, University Hospital Zurich, Zurich, Switzerland, 2 Institute of Medical Virology, University of Zurich, Zurich, Switzerland, 3 Department of Infectious Diseases, Bern University Hospital, University of Bern, Bern, Switzerland, 4 Division of Infectious Diseases and Hospital Epidemiology, Cantonal Hospital St. Gallen, Switzerland, 5 Sigma Research, London School of Hygiene and Tropical Medicine, United Kingdom, 6 Division of Infectious Diseases and Hospital Epidemiology, University Hospital Basel, University of Basel, Basel, Switzerland, 7 Basel Institute for Clinical Epidemiology, University Hospital Basel, Basel, Switzerland, 8 Division of Research and Analytical Services, Department of Informatics, University Hospital Basel, Basel, Switzerland, 9 Institute of Nursing Science, University of Basel, Basel, Switzerland, 10 Division of Infectious Diseases, Lausanne University Hospital, Lausanne, Switzerland, 11 Division of Infectious Diseases, Regional Hospital Lugano, Lugano, Switzerland

☯ These authors contributed equally to this work.
‡ AR and RDK also contributed equally to this work.
¶ Membership of the Swiss HIV Cohort Study (SHCS) is listed in the Acknowledgments.
* suraj.balakrishna@usz.ch

**Data Availability Statement:** The individual level datasets generated or analyzed during the current study do not fulfill the requirements for open data

## Abstract

Over the last decade, syphilis diagnoses among men-who-have-sex-with-men (MSM) have strongly increased in Europe. Understanding the drivers of the ongoing epidemic may aid to curb transmissions. In order to identify the drivers of syphilis transmission in MSM in Switzerland between 2006 and 2017 as well as the effect of potential interventions, we set up an epidemiological model stratified by syphilis stage, HIV-diagnosis, and behavioral factors to account for syphilis infectiousness and risk for transmission. In the main model, we used 'reported non-steady partners' (nsP) as the main proxy for sexual risk. We parameterized the model using data from the Swiss HIV Cohort Study, Swiss Voluntary Counselling and Testing center, cross-sectional surveys among the Swiss MSM population, and published syphilis notifications from the Federal Office of Public Health. The main model reproduced the increase in syphilis diagnoses from 168 cases in 2006 to 418 cases in 2017. It estimated that between 2006 and 2017, MSM with HIV diagnosis had 45.9 times the median syphilis incidence of MSM without HIV diagnosis. Defining risk as condomless anal intercourse with nsP decreased model accuracy (sum of squared weighted residuals, 378.8 vs. 148.3). Counterfactual scenarios suggested that increasing screening of MSM without HIV diagnosis and with nsP from once every two years to twice per year may reduce syphilis incidence (at most 12.8% reduction by 2017). Whereas, increasing screening among MSM with HIV

access: 1) The SHCS informed consent states that sharing data outside the SHCS network is only permitted for specific studies on HIV infection and its complications, and to researchers who have signed an agreement detailing the use of the data and biological samples; and 2) the data is too dense and comprehensive to preserve patient privacy in persons living with HIV. According to the Swiss law, data cannot be shared if data subjects have not agreed or data is too sensitive to share. Investigators with a request for selected data should send a proposal to the respective SHCS address (www.shcs.ch/contact). The provision of data will be considered by the Scientific Board of the SHCS and the study team and is subject to Swiss legal and ethical regulations and is outlined in a material and data transfer agreement. However, the numerical data underlying the study are presented in the main manuscript and supplementary information. The model's open-source code is available on https://github.com/Kouyos-Group/Suraj_syphilis_repository.

**Funding:** This study was funded by the Swiss HIV Cohort Study under the research grant to AR (Project 823) and by the Swiss National Science Foundation grants to AR (177499 and 324730_179567) and to RDK (BSSGI0_155851). None of the funders had a role in study design, data collection and analysis, decision to publish, or preparation of the manuscript.

**Competing interests:** I have read the journal's policy and the authors of this manuscript have the following competing interests: HFG, outside of this study, report grants from Swiss HIV Cohort Study, grants from Swiss National Science Foundation, grants from NIH, grants from Gilead unrestricted research grant, personal fees from Advisor/consultant for Merck, ViiV healthcare and Gilead Sciences and member of DSMB for Merck, grants from Yvonne Jacob Foundation. The institution of MC received research grants from Gilead, ViiV and MSD. All other authors declare no competing interests.

diagnosis and with nsP from once per year to twice per year may substantially reduce syphilis incidence over time (at least 63.5% reduction by 2017). The model suggests that reporting nsP regardless of condom use is suitable for risk stratification when modelling syphilis transmission. More frequent screening of MSM with HIV diagnosis, particularly those with nsP may aid to curb syphilis transmission.

## Author summary

Syphilis, one of the most common sexually transmitted infections, remains a major public health problem. Over the last decade, a rising number of diagnoses especially in men-who-have-sex-with-men (MSM) was observed in Western Europe and Northern America. In Switzerland, the number of syphilis diagnoses in MSM tripled between 2006 and 2017. In this study, we used a mathematical model to assess the drivers of this increase among MSM in Switzerland. Our model could reproduce the increase in syphilis diagnoses in both MSM with and without HIV diagnosis between 2006 and 2017. Based on this model we found that MSM with HIV diagnosis have an over 45 times higher syphilis incidence than MSM without HIV diagnosis. Furthermore, we found that reported sex with non-steady partners is a useful proxy of behavioral risk. Considering counterfactual scenarios, we showed that increasing the screening frequency for syphilis among MSM with HIV diagnosis and with non-steady partners from once a year to twice per year can reduce syphilis incidence by 63.5% to 99.2%.

## Introduction

Syphilis, a sexually transmitted infection caused by bacteria *Treponema pallidum*, remains a major public health problem. Over the last decade, a rising number of diagnoses especially in men-who-have-sex-with-men (MSM) was observed in Western Europe and Northern America [1–3]. In Switzerland, the number of syphilis diagnoses in MSM tripled between 2006 and 2017 [4]. A similar trend was observed for the incidence rate of syphilis in HIV-diagnosed MSM in the Swiss HIV Cohort Study (SHCS) [5–7]. Syphilis and HIV co-infections are common, and syphilis has been associated with an increased risk or HIV transmission [8]. Understanding syphilis transmission dynamics while considering the role of HIV infection could provide insights towards more effective prevention efforts to reduce the incidence of syphilis.

The syphilis incidence in a population is largely determined by parameters such as the transmission probability per sexual partnership, the average rate of acquisition of sexual partners and the duration of infectiousness, as well as by the prevalence of practices such as–oral sex, condomless anal intercourse, multiple sexual partnership, and anal intercourse with non-steady partners [9,10]. Prevention strategies, such as the promotion of condom use, risk-reduction counselling, and syphilis screening and treatment, have the potential to alter syphilis rates by modifying these key parameters [11]. Recent studies show a decrease in condom use for anal sex in MSM with HIV diagnosis [11,12]. This situation may have caused the increased incidence of syphilis.

The natural course of syphilis has four stages: primary, secondary, latent, and tertiary. Almost all syphilis transmissions occur through direct skin-to-skin contact with active lesions during the primary and secondary stages and the disease remains transmissible for an average period of two to three months [13]. Studies have estimated the attack rate of syphilis within 30 days of exposure to sexual transmission of syphilis to be between 16 and 30% [14,15]. During

the latent stage of syphilis, there are no clinical signs or symptoms of syphilis. If left untreated, the infection may reach its tertiary stage and may affect different organ systems. Hence it is essential to diagnose and treat people infected with syphilis as early as possible. A stepped wedge cluster randomized controlled trial showed that the routinized syphilis screening among men living with HIV increased the syphilis diagnosis [16]. Previous modelling studies suggest that more frequent syphilis screening could reduce incidence [17–20].

We aimed to identify possible drivers of the ongoing syphilis epidemic among MSM in Switzerland using a mathematical model (Fig 1) and to characterize the role of behavioral predictors (proxies for transmission risk of syphilis) that could best explain the changes in syphilis incidence observed over the last decade. Based on this model, we evaluated the effect of different screening strategies on syphilis incidence.

## Methods

### Ethics statement

The SHCS was approved by the local ethical committees of the participating centers: Kantonale Ethikkommission Zürich (KEK-ZH-NR: EK-793); Ethikkommission beider Basel ("Die Ethikkommission beider Basel hat die Dokumente zur Studie zustimmend zur Kenntnis genommen und genehmigt."); Kantonale Ethikkommission Bern (21/88); Comité départemental d'éthique des spécialités médicales es de médecine communautaire et de premier recours, Hôpitaux Universitaires de Genève (01–142); Commission cantonale d'éthique de la recherche sur l'être humain, Canton de Vaud (131/01); Comitato etico cantonale, Repubblica e Cantone Ticino (CE 813); Ethikkommission des Kantons St. Gallen (EKSG 12/003), and written informed consent was obtained from all participants.

We used data from the SHCS and cross-sectional surveys among the Swiss MSM population to estimate the transmission risk of syphilis. These temporal trends were used to parameterize a mathematical model, which was then used to assess the impact of different screening strategies via counterfactual scenarios (see Fig 1 for an overview of the workflow).

### Data

Our model population comprised all MSM in Switzerland from 2006 to 2018. We estimated the number of MSM in Switzerland between 2006 and 2018 based on the demographic dynamics of Switzerland by the Federal Statistical Office, Switzerland and Schmidt and Altpeter, 2019 [21,22]. In Switzerland, all syphilis cases are anonymously reported to the Federal Office of Public Health (FOPH) [4].

Approximately 80% of all HIV-diagnosed MSM in Switzerland are enrolled in the Swiss HIV Cohort Study (SHCS) [22,23]. The SHCS is a prospective multicenter cohort study, which systematically collects epidemiological and clinical data for more than 20000 HIV-diagnosed persons of age 16 or older in Switzerland since 1988 [24]. After the restart of routine syphilis screening in the SHCS in 2004, MSM are being tested for syphilis annually with treponemal and non-treponemal tests. We defined the first syphilis episode as a positive treponemal test after a negative treponemal test result. A $\geq$4-fold titer reduction or negativity in non-treponemal test result after an episode of syphilis indicate the absence of active syphilis. We defined subsequent episode(s) of syphilis as a subsequent $\geq$4-fold titer increase with a titer value of at least eight in non-treponemal test.

### Patterns of transmission risk

We estimated the proportion of MSM with HIV diagnosis ($MSM_{wHD}$) and with non-steady partners (nsP) based on the SHCS data. We estimated the yearly rates of switching from

**Step 1:** Establish the transmission model structure

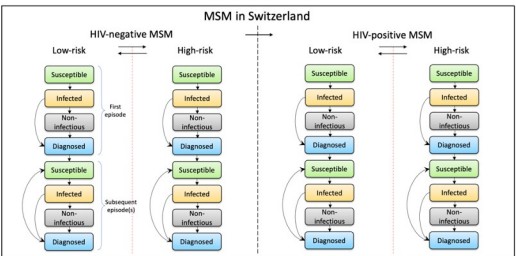

See Figure S1

**Step 2:** Parameterization of the model

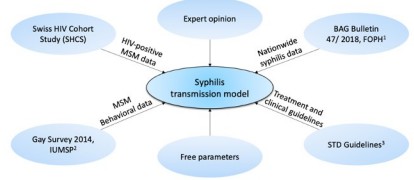

See section 3 of supplementary material

**Step 3:** Fit the transmission model to SHCS and syphilis notification data

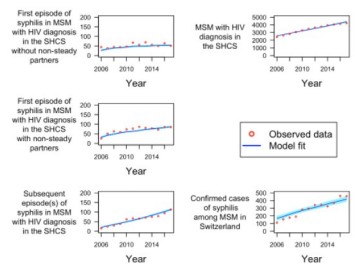

See Figure 2

**Step 4:** Test counterfactual scenarios

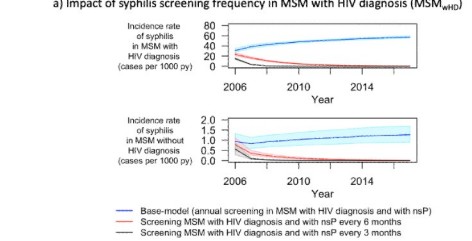

See Figure 4

**Step 5:** Perform sensitivity analyses to assess the robustness of the model

**Fig 1. Schematic overview of the methods.**

MSM$_{wHD}$ with nsP to MSM$_{wHD}$ without nsP and vice-versa using time-to-event survival analysis (section 6 in S1 Text). In contrast, we estimated an analogous proportion for MSM without HIV diagnosis (MSM$_{w/oHD}$) from published survey data (*Gaysurvey*) [25]. *Gaysurvey* was a series of anonymous internet-based self-administered surveys periodically conducted in

Switzerland among MSM since 1987. It was a part of the monitoring system established by the FOPH as a tool for monitoring risk behavior among MSM in Switzerland. In the last *Gaysurvey* in 2014, 834 people participated in the *Gaysurvey* and had a median age of 40 years. About 97% of these participants identified themselves either as gay/homosexual (84.1%) or bisexual (12.8%). As the data is not longitudinal at the individual level and is available for only a couple of years, a linear interpolation and a constant extrapolation was used to estimate the proportion of MSM with non-steady partners for $MSM_{w/oHD}$. As the participants of *Gaysurvey* were not followed up over time and hence participants could be different at each time point of survey conduction, we assumed a constant rate of switching back to $MSM_{w/oHD}$ with nsP from $MSM_{w/oHD}$ without nsP. We fixed this rate to the mean rate of switching back to $MSM_{wHD}$ with nsP from $MSM_{wHD}$ without nsP for the years 2006 and 2017. Based on this assumption, we estimated that the yearly rates of switching from $MSM_{w/oHD}$ without nsP to $MSM_{w/oHD}$ with nsP using a spline function (section 6 in S1 Text).

## Model of syphilis transmission

To model the syphilis epidemic in MSM in Switzerland between 2006 and 2017, we set up a system of 32 coupled ordinary differential equations (See section 2 in S1 Text for the model equations). We stratified the mathematical model by syphilis stage and cascade of care to account for infectiousness and disease progression into four levels: susceptible (MSM who do not have syphilis and can become infected with syphilis), infected (MSM who are infected with syphilis and can transmit; infectious), latent (MSM who are infected with syphilis but cannot transmit; non-infectious) and diagnosed (MSM diagnosed with syphilis) (Fig A in S1 Text). We did not include a 'treated' compartment as diagnosed individuals are assumed to be immediately treated and to clear from the infection within one month [13]. We further stratified the model by reported non-steady partners to account for increased transmission risk of syphilis. Additional layers in the model include HIV status ($MSM_{w/oHD}$ vs. $MSM_{wHD}$), the episode of syphilis: first (MSM who never had syphilis before) or subsequent (MSM who had at least one episode of syphilis before). Susceptible MSM become infected with syphilis with a force of infection ($\Lambda$) due to sexual contact with MSM infected with syphilis (section 3.3 in S1 Text). Infected MSM are infectious during the primary and secondary stages of syphilis that together last about 2–3 months after transmission. After this period, infected MSM enter the latent stage of syphilis during which the infection will no longer contribute to $\Lambda$. We further assume that infected and latently infected MSM would be diagnosed with syphilis if they are tested for syphilis and that all diagnosed MSM are immediately treated. Treated men would clear syphilis and become susceptible for subsequent episodes of syphilis. A complete description of the model, including model equations, is available in sections 1 and 2 in S1 Text.

Changes in transmission risk such as MSM with non-steady partners (nsP), as well as syphilis diagnosis and treatment rates among the $MSM_{wHD}$ were derived from the SHCS data, while changes in sexual risk behaviour among $MSM_{w/oHD}$ were derived from the *Gaysurvey*. Diagnosis rate and treatment rate of syphilis among the $MSM_{w/oHD}$ were derived from the literature, and Voluntary Counselling and Testing (VCT) data (Table B in S1 Text).

Parameters governing the transmission of syphilis and the syphilis incidence rate among the $MSM_{wHD}$ not enrolled in the SHCS were assumed to be the same as the $MSM_{wHD}$ enrolled in the SHCS. Parameters that could not be obtained from databases and literature such as transmission rate of syphilis, transmission rate of HIV, risk-sorting behaviour, and initial number of MSM infected with syphilis in 2006 were derived by fitting the model to data on syphilis cases among the $MSM_{wHD}$ and $MSM_{w/oHD}$ in Switzerland. This fit was done by minimizing the sum of squared weighted residuals between each observed and predicted datapoint

assuming normally distributed errors (section 7 in S1 Text). We further used simplified Markov chain Monte Carlo (MCMC), based on an adaptive Metropolis algorithm and including a delayed rejection procedure, with prior values specified by the distribution of the parameters to obtain their posterior distributions using the package "A Flexible Modelling Environment for Inverse Modelling, Sensitivity, Identifiability, and Monte Carlo Analysis (FME)" in R version 4.0.0 [26,27]. We used independent and uniform priors within *ad hoc* bounds for the free parameters (Table A in S1 Text). In a sensitivity analysis, we also fitted the model through maximization of the likelihood by assuming Poisson distributed incident cases of syphilis and compared the model fits.

## Counterfactual scenarios

We compared the incidence of syphilis obtained with the main model fit with that simulated when assuming the following two alternative scenarios. Firstly, we considered the impact of change in transmission risk (reported nsP in MSM) between the years 2006 and 2018 by assuming after 2006 no change in the proportion of MSM switching from with nsP to without nsP and vice-versa. Secondly, we considered the impact of change in screening frequency for syphilis between the years 2006 and 2018. We estimated the influence of this frequency by assuming an increase (by a given factor) of the screening frequency among $MSM_{wHD}$ and $MSM_{w/oHD}$ with non-steady partners.

## Sensitivity analysis

To test the robustness of the model and possible inconsistencies in data, we recalibrated the model in the following six ways: firstly, we used "reported condomless anal intercourse with non-steady partners" (nsCAI) as the proxy for sexual risk instead of nsP. Secondly, we assessed the impact of assuming the same transmission rate for $MSM_{wHD}$ and $MSM_{w/oHD}$ but taking into account a potential underreporting of syphilis cases among MSM without HIV diagnosis. Thirdly, we investigated the impact of a possible overestimation of transmission risk among MSM without HIV diagnosis. Fourthly, we assessed the impact of potential infectiousness of syphilis in the latent stage on the dependency between screening frequency and syphilis incidence. Fifthly, we tested the effect of model fitting through maximization of the likelihood by assuming Poisson distributed incident cases of syphilis. Finally, we tested the robustness of our model results to fixed parameter sets by sampling 50 sets of fixed parameters from given bounds around the point estimates using a Latin hypercube sampling algorithm. Sampling was followed by model optimization, and counterfactual scenario analyses.

## Results

### Data

Our mathematical model simulates transmission dynamics of syphilis in the MSM population in Switzerland from 2006 to 2018. Based on the Federal Statistical Office, Switzerland and Schmidt and Altpeter, 2019, we estimated the number of MSM in Switzerland in 2006 and 2018 to be 75016 and 85325, respectively (Fig B in S1 Text) [21,22]. There were 113 and 456 cases of syphilis reported to the FOPH in 2006 and 2018, respectively [4].

To estimate the number of MSM with HIV diagnosis ($MSM_{wHD}$) in Switzerland from 2006 and 2018 and the syphilis cases among them, we analyzed 5382 MSM enrolled in the Swiss HIV cohort study (SHCS) with a syphilis test between 2006 and 2018. The median registration year was 2007 [IQR (1999, 2013)] and the median year of birth was 1967 [IQR (1960, 1975)]. The key demographic characteristics of $MSM_{wHD}$ in the SHCS is summarized in Table 1. There were 2665

**Table 1. Characteristics of HIV-diagnosed MSM in the SHCS.**

| Variable | Value |
|---|---|
| Number of HIV-diagnosed MSM in the SHCS with at least one syphilis test between 2006 and 2018 | 5382 |
| Total person-years of follow-up between $1^{st}$ January 2006 and $31^{st}$ December 2018 | 35375.7 |
| Number of syphilis events observed during the observation period | 2165 |
| Median syphilis screening rate per person-year (IQR) | 1.20 (1.02 to 1.58) |
| White ethnicity, n (%) | 4855 (90.2) |
| Median year of birth (IQR) | 1967 (1960 to 1975) |
| Median age at HIV infection (IQR) | 35 (28 to 42) |
| Education level higher than high school, n (%) | 2484 (46.2) |
| Nadir CD4 (IQR) | 387.5 (272 to 533) |
| Ever used a recreational drug, n (%) | 1246 (23.2) |
| Alcohol consumption more than once a month for at least 6 months during the entire follow-up, n (%) | 4452 (82.7) |
| Current smoker, n (%) | 2598 (48.3) |
| Ever had a non-steady partner, n (%) | 3768 (70.0) |
| Ever had condomless anal intercourse with a non-steady partner, n (%) | 1942 (36.1) |

and 4366 MSM$_{wHD}$ enrolled in the SHCS in 2006 and 2018, respectively. By assuming that 80% of MSM$_{wHD}$ in Switzerland are enrolled in the SHCS, we estimated the number of MSM$_{wHD}$ in Switzerland in 2006 and 2018 to be 3031 and 5257 [22,23]. We also estimated syphilis cases reported among MSM$_{wHD}$ in Switzerland in 2006 and 2018 to be 106 and 312, respectively.

## Patterns of transmission risk

Information about reported non-steady partners and condomless anal intercourse was available for 5027 out of the analyzed 5382 MSM$_{wHD}$ (93.4%) enrolled in the SHCS during the observation period. 3768 out of 5027 MSM (75.0%) reported having a non-steady partner (nsP) at least once and 1942 (51.5%) among them reported condomless anal intercourse at least once. We estimated the proportion of MSM$_{wHD}$ with nsP to be 0.48 and 0.52 for the years 2006 and 2018, respectively (Fig C in S1 Text). We estimated that among MSM$_{wHD}$ the yearly rates of switching from without nsP to with nsP has decreased only slightly from 0.49 in 2006 to 0.41 in 2017. For the same period, we estimated that the yearly rates of switching back from with nsP to without nsP has decreased from 0.74 in 2006 to 0.45 in 2017 (Fig D in S1 Text). We estimated the proportion of MSM$_{w/oHD}$ with nsP to be 0.74 and 0.73 for the years 2006 and 2018 respectively using data published in *Gaysurvey* (Fig E in S1 Text) [25]. Our results were similar to the proportion of MSM$_{w/oHD}$ with nsP (0.75) published in European Men-who-have-sex-with-men Internet Survey (EMIS) 2017 [28]. As the *Gaysurvey* data is not longitudinal, we assumed a constant rate of switching for MSM$_{w/oHD}$ from without nsP to with nsP. It equaled the mean rate of switching for MSM$_{wHD}$ from without nsP to with nsP for the years 2006 and 2017 and estimated it to be 0.54. Based on the *Gaysurvey* data, we then estimated that the yearly rate of switching for MSM$_{w/oHD}$ from without nsP to with nsP has increased by 2.7% (1.44 in 2006 vs. 1.48 in 2017) (Fig F in S1 Text). Detailed information on the data used in our model is provided in the supplementary material (Tables A and B in S1 Text).

## Observed and modelled data

The model could accurately reproduce the number of MSM$_{wHD}$ in Switzerland and the syphilis cases among MSM$_{wHD}$ and MSM$_{w/oHD}$ (Fig 2). The model estimated that the number of new

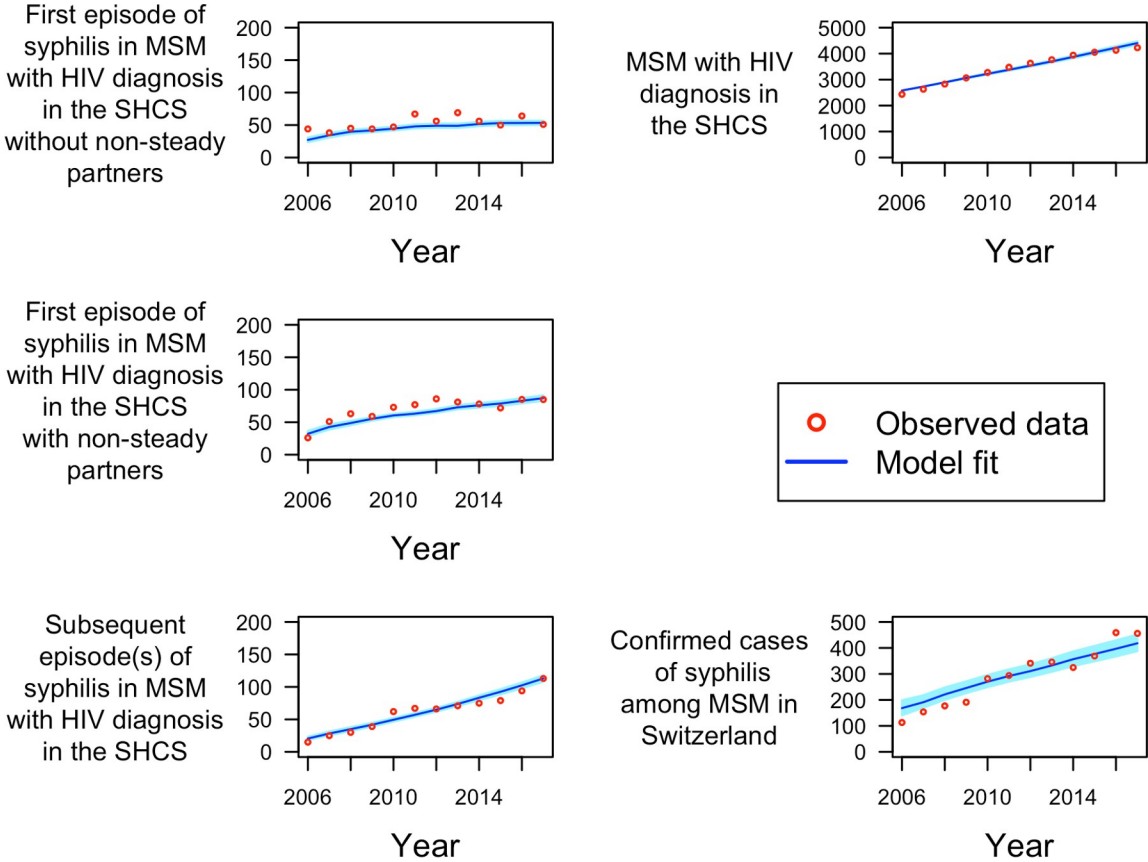

**Fig 2. Model fit;** MSM: men who have sex with men. SHCS: Swiss HIV Cohort Study; Red circles represents the observed datapoints obtained from the SHCS and literature. Blue line represents the median of the simplified Markov chain Monte Carlo (MCMC) model fit. Light blue shaded region represents the 95% quantile of the MCMC model fit.

syphilis infections among MSM for the years 2006 to 2017 ranged between 168 and 418 diagnosed cases per year, which corresponds to an overall incidence rate of 2.24 and 4.91 syphilis cases per 1000 person-years (median syphilis incidence rate of 3.77 cases per 1000 person-years).

In addition, the model provided an estimate for the incidence rate of syphilis among $MSM_{wHD}$ and $MSM_{w/oHD}$ stratified by reported non-steady partners (Fig 3). $MSM_{wHD}$ had 45.93 [IQR (45.28, 46.32)] times the incidence of $MSM_{w/oHD}$, and MSM with non-steady partners had 1.63 [IQR (1.60, 1.67)] times the incidence of MSM without non-steady partners. Characterization of model optimization, model performance, and sampling for uncertainty estimates are depicted in the supplementary material (Figs G-K and Table C in S1 Text)

## Counterfactual scenarios

We compared the incidence rate of syphilis obtained with the model fitted to the observed number of syphilis cases with simulations assuming the following alternative scenarios: firstly, we considered the impact of change in transmission risk between the years 2006 and 2018. With no change in the proportion of MSM switching from MSM with nsP to MSM without nsP and vice-versa between the years 2006 and 2018, we estimated a reduction of 17.56% (4.04 vs. 4.91 cases per 1000 person-years) in the incidence rate of syphilis in 2017 compared to incidence rate obtained by the model fit in 2017 (Fig L in S1 Text). Secondly, we considered the impact of change in screening frequency for syphilis between the years 2006 and 2018.

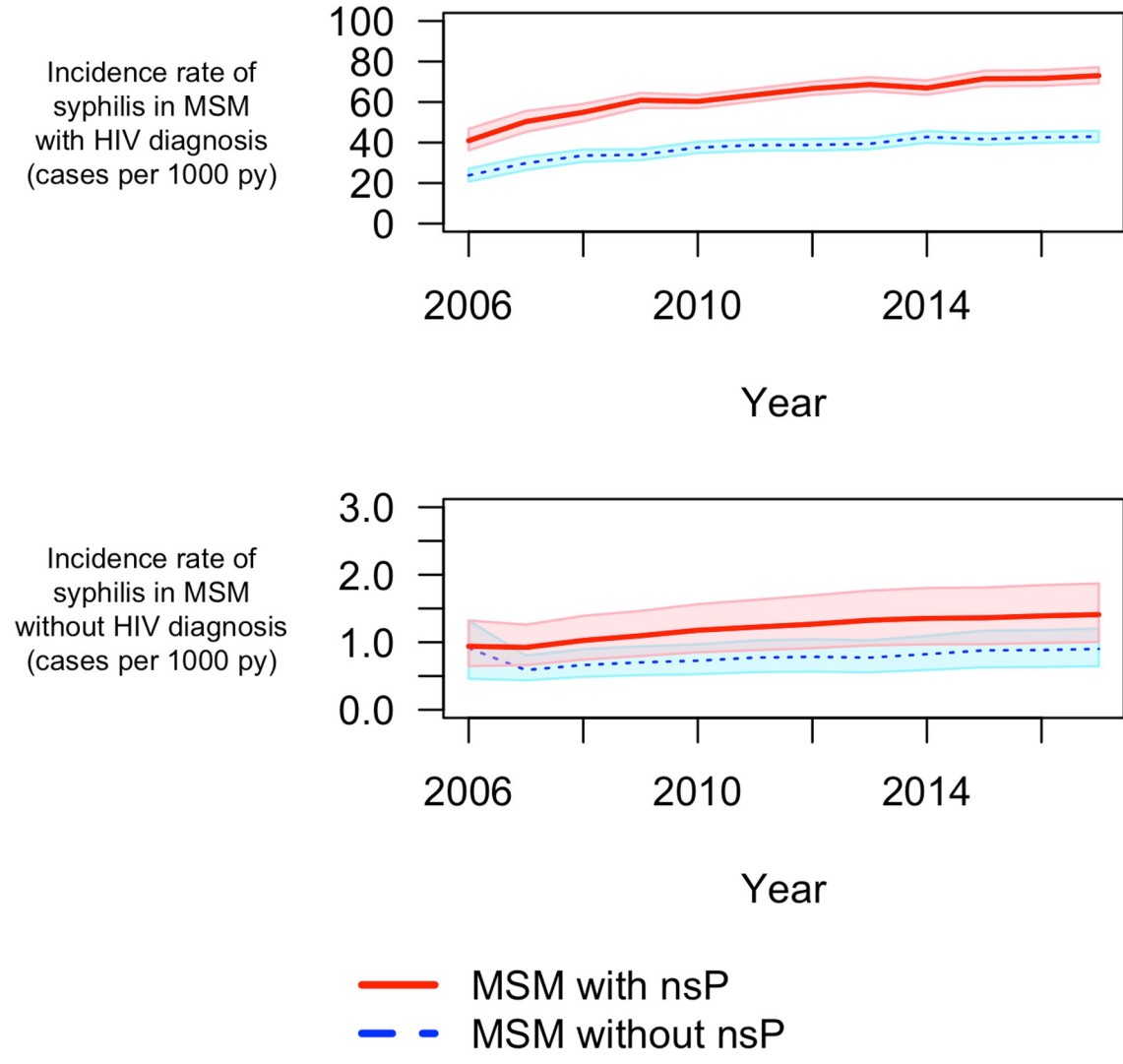

**Fig 3. Model simulations–Incidence of syphilis stratified by transmission risk (reported non-steady partners) and HIV-status;** MSM–men who have sex with men; nsP–non-steady partners; py–person-years; Solid red and dashed blue lines represent incidence rate of syphilis in MSM with and without nsP, respectively. The shaded regions represent the 95% quantile for the respective incidence rates.

Increasing the frequency of screening for syphilis among MSM with HIV diagnosis and with nsP from once per year to twice per year during the observation period (corresponding to 48,576 syphilis screening tests instead of 24,179 tests), we estimated a reduction of 99.19% (0.04 vs. 4.91 cases per 1000 person-years) in the incidence rate of syphilis in 2017 compared to incidence rate obtained by the model fit in 2017. By contrast, we estimated only a reduction of 12.75% (4.28 vs. 4.91 cases per 1000 person-years) in the incidence rate of syphilis in 2017 by increasing the frequency of screening for syphilis among MSM without HIV diagnosis and with nsP from once per two years to twice per year during the observation period (corresponding to 1,289,070 syphilis screening tests instead of 332,226 tests) (Fig 4).

## Sensitivity analysis

To test the robustness of the model and possible inconsistencies in data, we recalibrated the model and determined the corresponding change in the incidence rate of syphilis, the

## a) Impact of syphilis screening frequency in MSM with HIV diagnosis (MSM$_{wHD}$)

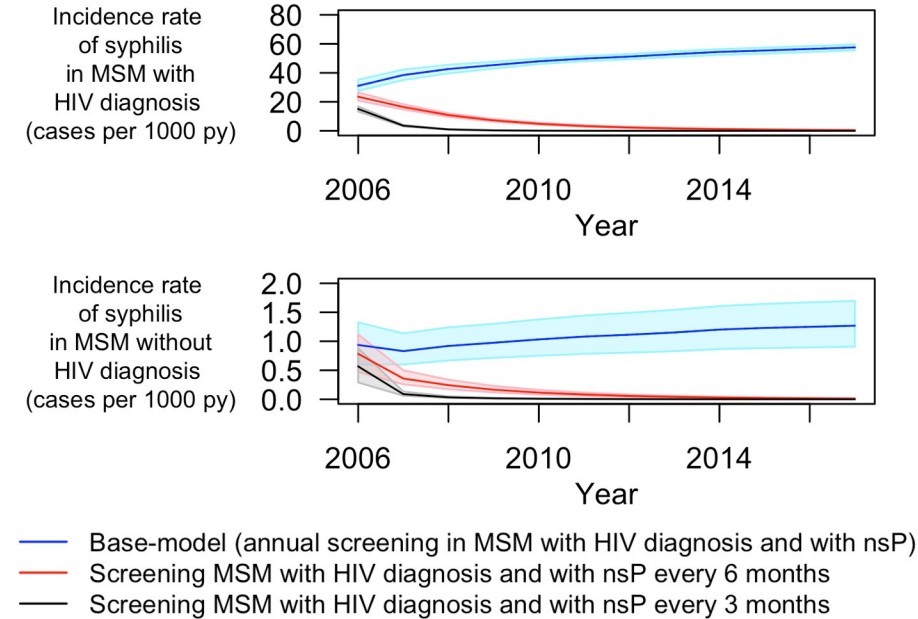

Base-model (annual screening in MSM with HIV diagnosis and with nsP)
Screening MSM with HIV diagnosis and with nsP every 6 months
Screening MSM with HIV diagnosis and with nsP every 3 months

## b) Impact of syphilis screening frequency in MSM without HIV diagnosis (MSM$_{w/oHD}$)

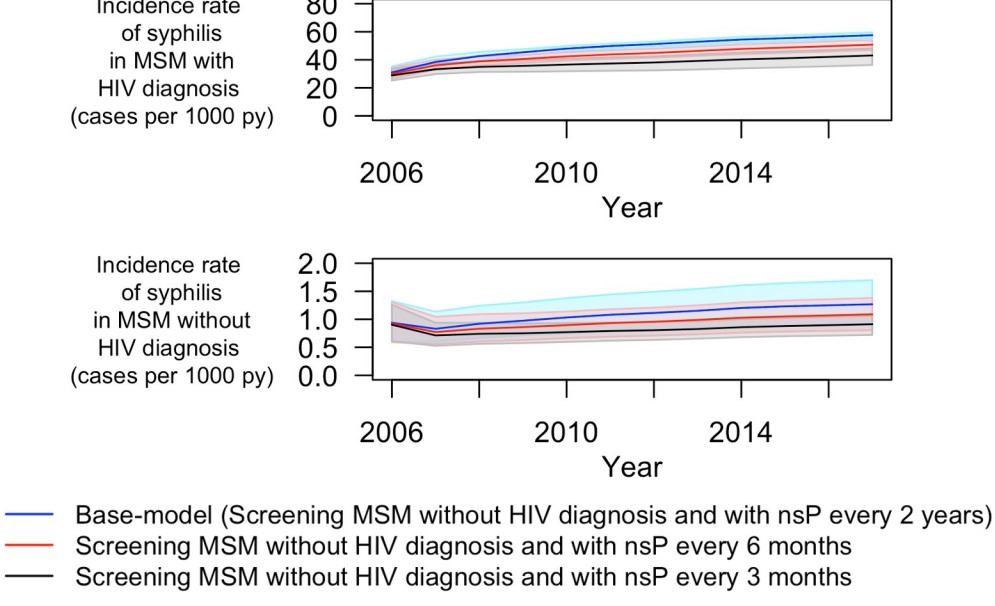

Base-model (Screening MSM without HIV diagnosis and with nsP every 2 years)
Screening MSM without HIV diagnosis and with nsP every 6 months
Screening MSM without HIV diagnosis and with nsP every 3 months

**Fig 4. Counterfactual scenario: impact of change in screening frequency for syphilis.** MSM–men who have sex with men; nsP–non-steady partners; py–person-years; Blue lines represent incidence rate of syphilis in the base model (initial model fit). Red and black lines in panel a) represent incidence rate of syphilis that obtained for the counterfactual scenarios when MSM with HIV diagnosis were screened for syphilis every 6 and 3 months instead of once a year, respectively. Red and black lines in panel b) represent incidence rate of syphilis that obtained for the counterfactual scenarios when MSM without HIV diagnosis were screened for syphilis every 6 and 3 months instead of once every 2 years, respectively. The shaded regions represent the 95% quantile for the respective incidence rates.

fitted parameters, and the goodness of fit (sum of squared weighted residuals) (Table D in S1 Text).

Firstly, we considered alternative criteria to stratify transmission risk (Fig M in S1 Text). When assuming reported non-steady condomless anal intercourse (nsCAI) as the transmission risk instead of nsP, we estimated that MSM with nsCAI had 1.71 [IQR (1.65, 1.81)] times the incidence of MSM without nsCAI. However, the goodness of fit for this model was considerably worse than for the main model (378.8 vs. 148.3). Secondly, we assessed the impact of the assumption of different transmission rates of syphilis among $MSM_{wHD}$ and $MSM_{w/oHD}$. In the main model we had to assume different transmission rates, because of an apparent discrepancy in the data: despite similar frequency of reported nsP among $MSM_{wHD}$ and $MSM_{w/oHD}$, we observed much higher incidence of syphilis among $MSM_{wHD}$ than $MSM_{w/oHD}$ even when taking serosorting into account (Table B in S1 Text). To resolve this contradiction, in the main model, we assumed different transmission rates of syphilis (beta) in $MSM_{wHD}$ and $MSM_{w/oHD}$ in our model (Table A in S1 Text). In the absence of this assumption, the model was unable to reproduce the syphilis epidemic in Switzerland and led to either a large overestimation of syphilis incidence in $MSM_{w/oHD}$ or a large underestimation in $MSM_{wHD}$ (Fig N in S1 Text). As an alternative approach to capture the discrepancy in the data, we fitted a model assuming that there is no difference in transmission rate of syphilis among $MSM_{wHD}$ and $MSM_{w/oHD}$ but instead including a parameter to account for possible underreporting of syphilis cases to FOPH among $MSM_{w/oHD}$ (Fig O in S1 Text). However, the goodness of fit for this model was worse than for the main model (sum of squared weighted residuals, 206.6 vs. 148.3) and required the assumption of an unrealistically high underreporting factor (40.8). Thirdly, we investigated the impact of a possible overestimation of transmission risk among $MSM_{w/oHD}$. We fitted models assuming the proportion of $MSM_{w/oHD}$ with nsP to be 25%, 50%, and 75% lower than the proportion of $MSM_{w/oHD}$ with nsP estimated from data published in the Gay-survey (Fig P in S1 Text). This corresponds to proportion of $MSM_{w/oHD}$ with nsP in 2017 to be 0.54, 0.36, and 0.18 vs. 0.73, respectively. The estimated goodness of fit values for these models were slightly better than the goodness of fit for the main model (136.4, 124.4, and 123.6 vs. 148.3 respectively) suggesting that model is robust to the hypothesized overestimation.

Fourthly, we assessed the impact of the duration of syphilis infectiousness on the dependency between the screening frequency and the syphilis incidence. Instead of assuming zero infectiousness in the latent stage of syphilis (>3 months since infection), we varied the infectiousness of syphilis in the latent stage to be 1% and 10% of that in the primary and/or secondary stage of syphilis ($\leq$ 3 months since infection) (Fig Q in S1 Text). The goodness of fit values for these models were worse than for the main model (154.3 and 397.6 vs 148.3 respectively). As depicted above in the counterfactual scenario, increase in the frequency of screening for syphilis among $MSM_{wHD}$ with nsP led to reduced simulated syphilis incidence in 2017. Increased infectiousness during the latent stage led to a smaller estimated reduction in syphilis incidence in 2017 in the counterfactual scenarios (93.59% and 63.50% vs. 99.19% by assuming the infectiousness in latent stage of syphilis to be 1% and 10% vs. 0% of that in primary and secondary stage of syphilis respectively) (Fig 5)

Fifthly, we tested the effect of using an alternative fitting procedure. Instead of fitting the model through minimization of the sum of squared weighted residuals, we fitted the model through maximization of the likelihood by assuming Poisson distributed incident cases of syphilis. We obtained similar median incidence rate and model fit as obtained by the main approach (Fig R in S1 Text).

Finally, we tested the robustness of our model by assessing if the model results were reliant on the choice of value of fixed parameters by sampling fixed parameter sets using Latin hypercube sampling followed by model optimization, and counterfactual scenario analyses (section

## a) Impact of syphilis screening frequency for assuming 1% infectiousness in latent stage of that in primary and secondary stage of syphilis

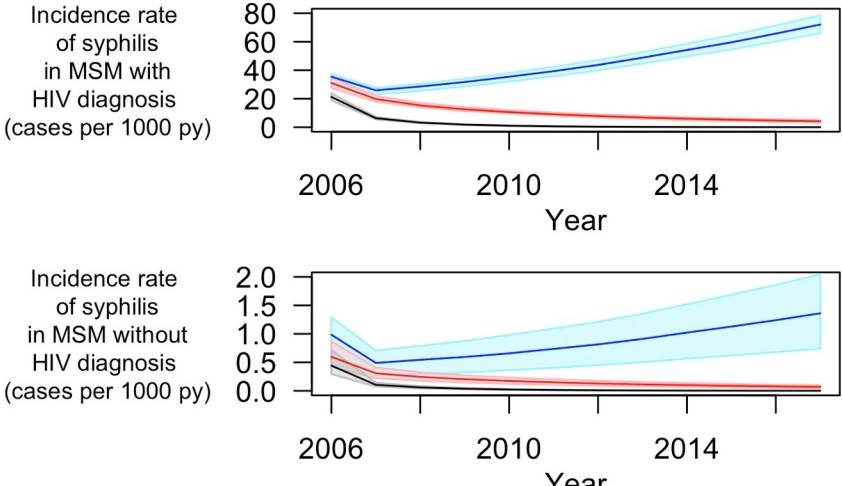

## b) Impact of syphilis screening frequency for assuming 10% infectiousness in latent stage of that in primary and secondary stage of syphilis

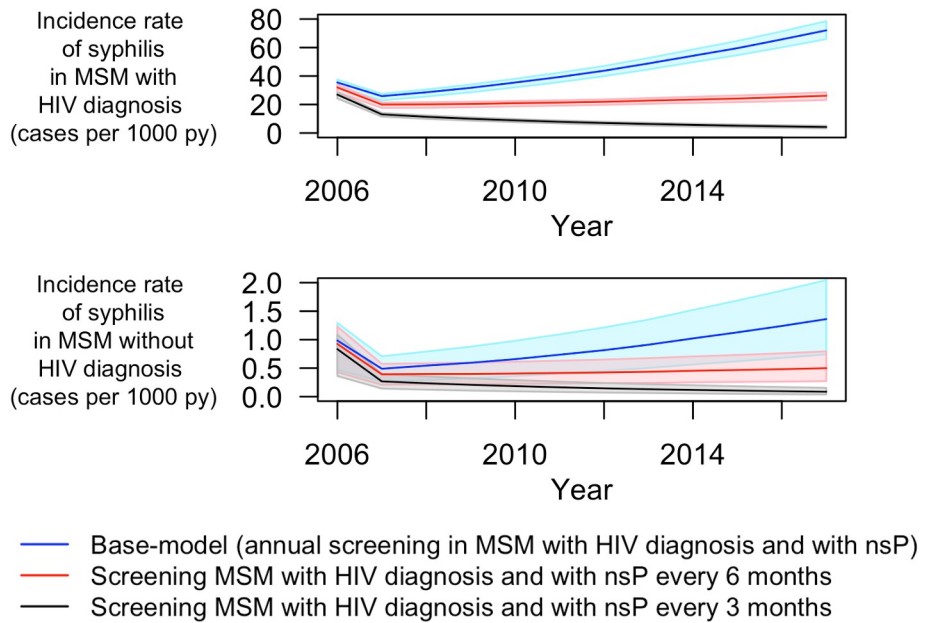

—— Base-model (annual screening in MSM with HIV diagnosis and with nsP)
—— Screening MSM with HIV diagnosis and with nsP every 6 months
—— Screening MSM with HIV diagnosis and with nsP every 3 months

**Fig 5. Sensitivity analysis: impact of allowing for syphilis transmission to occur in the latent stage.** Panel a) and b) show the impact of syphilis screening frequency on syphilis incidence assuming the infectiousness of syphilis in the latent stage to 1% and 10% of that in primary and secondary stage of syphilis, respectively; MSM–men who have sex with men; nsP–non-steady partners; py–person-years; Blue lines represent incidence rate of syphilis in the base model (initial model fit). Red and black lines represent incidence rate of syphilis that obtained for the counterfactual scenarios when MSM with HIV diagnosis were screened for syphilis every 6 and 3 months instead of once a year, respectively. The shaded regions represent the 95% quantile for the respective incidence rates.

11 in S1 Text). We found that our model results were robust to the choice of fixed parameter sets (Figs S and T in S1 Text).

## Discussion

The transmission model we present was able to accurately reproduce the syphilis epidemic in Switzerland. Our study outlines the much higher incidence of syphilis in MSM with HIV diagnosis compared to MSM without HIV. In our simulations, MSM with HIV diagnosis have 46 times the syphilis incidence of MSM without HIV diagnosis. On average, MSM with non-steady partners (nsP) have 1.6 times the syphilis incidence in MSM without nsP. Also, constant yearly rates of switching from MSM without nsP to MSM with nsP (our measure of transmission risk) and vice-versa during the observation period would have led to a lower incidence of syphilis in 2017 by 17.6%. Finally, increasing the screening frequency of syphilis to twice per year in MSM with HIV diagnosis and with nsP would have contributed to a much higher reduction in syphilis incidence than increasing the screening frequency to twice per year in MSM without HIV diagnosis and with nsP (99.2% vs. 12.8%).

A key challenge in simultaneously modelling syphilis transmission in $MSM_{wHD}$ and $MSM_{w/oHD}$ population is the apparent discrepancy in the data: when comparing the data from the FOPH and the SHCS, $MSM_{wHD}$ exhibited 46 times the syphilis incidence of $MSM_{w/oHD}$, even though proportions of $MSM_{w/oHD}$ with nsP obtained from the data published in *Gaysurvey* was higher than the proportion of $MSM_{wHD}$ with nsP estimated from the SHCS. To resolve this contradiction, we had to assume different transmission rates of syphilis in $MSM_{wHD}$ and $MSM_{w/oHD}$ in our model. Based on fitting the model, we estimated a greater than 20-fold in transmission rate of syphilis in $MSM_{wHD}$ compared to $MSM_{w/oHD}$, even after adjusting for contact probabilities and difference in screening frequency among $MSM_{wHD}$ and $MSM_{w/oHD}$. As the majority of the $MSM_{wHD}$ in Switzerland are virally suppressed, it is unclear whether a biological explanation for a higher transmission rate of syphilis in $MSM_{wHD}$ is plausible. Alternatively, this difference may also be due to an unadjusted transmission risk of syphilis such as the number of partners and frequency of contacts, which we did not explicitly model.

To explore alternative distributions of risk, we stratified the structure of transmission risk in our model based on reported condomless anal intercourse with non-steady partners (versus reported non-steady partners). The model fit was worse than that in the main analysis. This might be because syphilis could be transmitted through oral sex, and using condoms for anal intercourses is not completely protective against syphilis [29]. In an alternative approach, we assumed no difference in transmission rate of syphilis among $MSM_{wHD}$ and $MSM_{w/oHD}$ and allowed the model to account for possible underreporting of syphilis cases to FOPH among $MSM_{w/oHD}$ to explain the apparent difference in the incidence rate among $MSM_{wHD}$ and $MSM_{w/oHD}$. We estimated the incidence of syphilis in $MSM_{w/oHD}$ to be 40.8 times that reported by FOPH. This hypothesized underreporting of syphilis cases, although not in the same magnitude, seems to be supported by the European Men-who-have-sex-with-men Internet Survey (EMIS) 2017 [28]. Based on the EMIS-2017 survey, 3.4% of all European $MSM_{w/oHD}$ and 13.9% of all European $MSM_{wHD}$ (overall 4.4%) were diagnosed syphilis in the previous twelve months. If the EMIS-2017 were representative of MSM in Switzerland, the number of syphilis cases would have been about 6.3 times those reported to the FOPH. However, about 70% of the participants of EMIS-2017 were recruited through geosocial dating apps such as Planet-Romeo, Grindr, and Hornet. Several studies have shown that MSM who use geosocial dating apps have higher number of partners, frequency of unprotected sex, and incidence of sexually transmitted diseases compared to MSM who do not use such apps [30–33]. Hence,

the incidence of syphilis among participants of EMIS 2017 is likely higher than that in the overall MSM population in Switzerland.

Persons infected with syphilis are usually symptomatic during the primary and secondary stage of syphilis that lasts about 2–3 months since infection. After this period, individuals enter latent stage of syphilis during which they become asymptomatic, yet still detectable through syphilis screening. If left untreated, individuals can develop late/tertiary syphilis disease any time from 1 to 30 years after primary infection. About 25–40% of individuals with untreated syphilis can develop such diseases [34]. Hence, $MSM_{w/oHD}$ may be infected with syphilis but are not diagnosed due to lower frequency of syphilis screening and eventually may have non-syphilis related deaths, resulting in an underreporting of syphilis cases to FOPH. Moreover, despite syphilis being a notifiable disease, it is possible that not all cases are reported to the FOPH resulting in underreporting. However, the extent of underreporting is not clear and warrants further investigation.

Also, there is no clear evidence of the infectiousness of syphilis in the latent stage of syphilis (>3 months since infection) [13]. We tested the influence of infectiousness in the latent stage of syphilis. Our model could reproduce the syphilis epidemic for these scenarios and estimated similar incidence rates compared to the main model fit. By means of counterfactual scenarios, we further assessed the impact of infectiousness of syphilis in latent stage on the dependency between the screening frequency and the syphilis incidence. We found that the estimated reduction in syphilis incidence due to increase in screening frequency decreases when assuming residual infectiousness during the latent stage. Infectiousness during the latent stage of syphilis warrants further investigation as it may have an impact on the effectiveness of different screening strategies. However, even when we assumed infectiousness during the latent stage to be as high as 10% of that in primary and secondary stage, we observed 63.5% reduction in syphilis incidence in 2017 when screening of syphilis was performed twice per year instead of once per year in $MSM_{wHD}$ with nsP. Our study results thus robustly indicate that frequent screening of MSM with HIV diagnosis and with nsP is required to curb syphilis transmission. This supports the EACS guidelines which recommend to "consider more frequent than annual screening if at risk" [35].

As mentioned before, we observed that the proportion of $MSM_{w/oHD}$ with non-steady partners is higher than the proportion of $MSM_{wHD}$ with non-steady partners. On one hand, $MSM_{wHD}$ in the SHCS answer the questionnaires on sexual behaviour during consultation or interviews with physicians or nurses they are likely to have recurrent contact with, while enrolled in the cohort. This may cause social desirability bias, thereby resulting in potential underestimation of transmission risk among $MSM_{wHD}$. On the other hand, MSM participating in *Gaysurvey* may not be representative of MSM in Switzerland, as MSM with nsP may be more likely to enroll in online surveys such as *Gaysurvey* compared to MSM without nsP, thereby resulting in an overestimation in transmission risk among $MSM_{w/oHD}$. We tested the influence of overestimation in transmission risk among $MSM_{w/oHD}$ estimated from data published from *Gaysurvey*. Our model could reproduce the syphilis epidemic for these scenarios and estimated similar incidence rates and similar goodness of fit compared to the main model fit indicating the robustness of our model. Finally, we fitted the model using maximization of likelihood by assuming Poisson distributed incident cases of syphilis and found similar results to that obtained by main model fit.

Our transmission model enabled us to integrate data from different sources such as published data from the FOPH and the literature, and the results estimated from the SHCS and cross-sectional studies to assess the drivers of the syphilis epidemic in Switzerland. Previous studies have shown the potential impact of more frequent screening in MSM for syphilis on reducing syphilis infections [17–20]. An individual-based mathematical model that simulates

the formation and breakup of sexual partnerships and tracks the transmission of syphilis within a synthetic population of sexually active gay men showed that increasing the frequency of syphilis screening can have a large impact on reducing syphilis prevalence [17]. The study also emphasized targeted frequent screening of gay men who have large numbers of partners or who engage in group sex to be a more efficient way of reducing syphilis transmission. An agent-based, network model of syphilis transmission, representing a core population of 2000 Canadian MSM at particular high risk for syphilis, explicitly estimated the effect of different screening frequencies on the projected annual rates of syphilis in this population [18]. The study showed that increased syphilis screening frequency was consistently more effective than improved coverage for screening. Our model differs from these studies, in that we not only focused on a core group of MSM by means of transmission risk of syphilis but also stratified the model based on HIV status. To the best of our knowledge, this is the first study to link the syphilis epidemic in $MSM_{wHD}$ with that in $MSM_{w/oHD}$ and to estimate the effect of screening strategies implemented in target groups of the population on the overall epidemic.

One of the main limitations of our study is the uncertainty regarding the representativeness of the data used for $MSM_{w/oHD}$. This resulted in model assumptions which required critical sensitivity analyses to assess the robustness of our findings. Our study has however a substantial strength regarding $HIV_{wHD}$: the use of routine serological screening in a representative population over many years. The estimation of transmission risk of syphilis in $MSM_{wHD}$ was based on the SHCS data. As the SHCS is highly representative for $MSM_{wHD}$ in care in Switzerland, we expect these data to provide accurate estimates. In contrast, estimation of transmission risk in $MSM_{w/oHD}$ was based on cross-sectional studies which might have led to more uncertainty in our results. The degree of mixing between $MSM_{wHD}$ and $MSM_{w/oHD}$ was based on the estimated frequency of HIV-discordant status in the data collected from VCT checkpoints across Switzerland where people can get spontaneously tested for sexually transmitted infections in a fully anonymized way. As bias towards screening upon transmission risk is expected, representativeness cannot be warranted. We stratified the model based on reported non-steady partners and condomless anal intercourse with non-steady partners as the transmission risk of syphilis. Although the available data in the SHCS did not allow us to account for the effect of other risk factors such as number of partners and frequency of sexual acts, the goodness of fit suggests reported non-steady partner to be a good proxy for risk stratification to model syphilis transmission.

In conclusion, our model reconstructed the syphilis epidemic among MSM in Switzerland by capturing changes in sexual risk behaviour and interactions between HIV MSM with and without HIV diagnosis. It suggests that having non-steady partners (nsP) regardless of condomless anal intercourse would be a suitable criterion for risk stratification when modelling syphilis transmission. More frequent screening and treating MSM with HIV diagnosis, particularly those with nsP may aid to curb syphilis transmission.

## Supporting information

**S1 Text. Supplementary information.**
(PDF)

## Acknowledgments

We thank the patients for participating in the SHCS, the study nurses, physicians, data managers, and the administrative assistants.

The members of the SHCS are Anagnostopoulos A, Battegay M, Bernasconi E, Böni J, Braun DL, Bucher HC, Calmy A, Cavassini M, Ciuffi A, Dollenmaier G, Egger M, Elzi L, Fehr J, Fellay J, Furrer H, Fux CA, Günthard H (President of the SHCS), Haerry D (deputy of "Positive Council"), Hasse B, Hirsch HH, Hoffmann M, Hösli I, Huber M, Kahlert CR (Chairman of the Mother and Child Substudy), Kaiser L, Keiser O, Klimkait T, Kouyos RD, Kovari H, Ledergerber B, Martinetti G, Martinez de Tejada B, Marzolini C, Metzner KJ, Müller N, Nicca D, Paioni P, Pantaleo G, Perreau M, Rauch A (Chairman of the Scientific Board), Rudin C, Scherrer AU (Head of Data Centre), Schmid P, Speck R, Stöckle M (Chairman of the Clinical and Laboratory Committee), Tarr P, Trkola A, Vernazza P, Wandeler G, Weber R, and Yerly S.

## Author Contributions

**Conceptualization:** Suraj Balakrishna, Luisa Salazar-Vizcaya, Axel J. Schmidt, Andri Rauch, Roger D. Kouyos.

**Data curation:** Suraj Balakrishna, Katharina Kusejko.

**Formal analysis:** Suraj Balakrishna, Luisa Salazar-Vizcaya.

**Funding acquisition:** Andri Rauch, Roger D. Kouyos.

**Investigation:** Suraj Balakrishna, Luisa Salazar-Vizcaya, Roger D. Kouyos.

**Methodology:** Suraj Balakrishna, Luisa Salazar-Vizcaya, Axel J. Schmidt.

**Project administration:** Axel J. Schmidt.

**Resources:** Axel J. Schmidt, Maria Christine Thurnheer, Huldrych F. Günthard.

**Software:** Suraj Balakrishna, Viacheslav Kachalov.

**Supervision:** Andri Rauch, Roger D. Kouyos.

**Validation:** Suraj Balakrishna, Luisa Salazar-Vizcaya.

**Visualization:** Suraj Balakrishna, Luisa Salazar-Vizcaya.

**Writing – original draft:** Suraj Balakrishna.

**Writing – review & editing:** Luisa Salazar-Vizcaya, Axel J. Schmidt, Viacheslav Kachalov, Katharina Kusejko, Maria Christine Thurnheer, Jan A. Roth, Dunja Nicca, Matthias Cavassini, Manuel Battegay, Patrick Schmid, Enos Bernasconi, Huldrych F. Günthard, Andri Rauch, Roger D. Kouyos.

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
