## [Decision Letter · Decision Letter 0]

3 Jun 2021

Dear Mr Balakrishna,

Thank you very much for submitting your manuscript "Assessing the drivers of syphilis among men who have sex with men in Switzerland reveals a key impact of testing frequency: A modelling study" for consideration at PLOS Computational Biology.

As with all papers reviewed by the journal, your manuscript was reviewed by members of the editorial board and by several independent reviewers. In light of the reviews (below this email), we would like to invite the resubmission of a significantly-revised version that takes into account the reviewers' comments.

We cannot make any decision about publication until we have seen the revised manuscript and your response to the reviewers' comments. Your revised manuscript is also likely to be sent to reviewers for further evaluation.

Sincerely,

Benjamin Muir Althouse

Associate Editor

PLOS Computational Biology

Thomas Leitner

Deputy Editor

PLOS Computational Biology

Editor comments:

Please fully address reviewer 2's concerns over communication of the model and the results. It is hard to evaluate the modeling methods without understanding the model.

Reviewer's Responses to Questions

**Comments to the Authors:**

Reviewer #1: 1. The introduction is very shallow. The authors should do more by reviewing existing literature in this area.

2. The objective of this study must be clearly stated from the onset

3. The authors talked about system of nonlinear equations for the study. This I could not find. Authors must include the model equations.

4. The results of the study must relate with previous literature and authors must state how this model or result improved on previous results.

5. The fonts of the figures must be increased

6. The authors should go through the entire work to check for typo errors

Reviewer #2: Though I’ve spent substantial time with the main manuscript and supplemental materials for the paper titled “Assessing the drivers of syphilis among MSM in Switzerland reveals a key impact of testing frequency: a modelling study”, my understanding of this work is still incomplete. The authors are using a fairly complicated ODE model involving syphilis transmission, HIV demographics, heterogenous sexual activity classes, and history of syphilis infection. They use multiple sources of data for both parameterization and model fitting. They assess a few counterfactual intervention scenarios and conclude that increased screening among HIV+ MSM who report non-steady partners would be most effective, compared to increased screening of other groups. The work suggests that if syphilis screening among MSM with HIV were quadrupled (from once every other year to twice per year), syphilis incidence would drop to 0.04 per year from 4.91 per year—a drastic reduction that appears to lead to syphilis elimination in this modeled population, though this is not stated directly.

My main concerns involve elements of the methods and supplemental sections related to a) model construction, b) model parameterization, and c) model communication. The communication issues are the most serious, as I’m unable to assess whether critical model features are implemented in a suitable way.

Model construction:

1) Line 152 refers to 36 equations. Subsequent text (152-162) and Figure 1a would indicate only 32 equations (4 syphilis states, 2 HIV states, 2 nSP states, 2 syph_history states). What do the 4 remaining equations refer to?

2) HIV prevalence was not stable between 2006-2018 (line 236 and Supplemental Table 2): the prevalence grew from 3.5% in 2006 to 5.2% in 2018). It seems the serosorting parameters are problematic, as the probability of a contact being with an HIV+/- person will in part depend on the changing density of HIV+/- people. Otherwise there is changing preference for HIV+ partners over the period as the prevalence changes.

3) Critical: Tau (the relative risk due to reported non steady partners) does not appear to be described in the supplemental section 3.3. This seems like a critical parameter that is specified (i.e., not free). Can this be thought of as a contact rate inflation factor?

Model parameterization:

4) The data specified in supplemental table 2 all seem like point estimates with no variability. Is this wise? For example, do we really know the serosorting probability with exact precision? Perhaps this should be varied. I understand you don’t want to have an infinite number of “free parameters”, but perhaps having partially informed group would be a good idea. Your inferences might be quite reliant on the choice of values for this serosorting parameter (and others).

Model communication:

5) Critical: Screening versus testing versus diagnosis. The supplementals give “diagnosis rates” in Table 2. The primary counterfactual intervention is increased screening. This language needs to be harmonized throughout. There are subtle differences between rates of screening, testing, and diagnosis.

6) As a reviewer it is critical for me to understand how screening and testing work in the model, as this is the primary intervention being proposed. From Section 3.4 in the supplemental; I’m trying to understand delta_LD and delta_ID, but I am confused. From the section text: “the rates of becoming diagnosed from infected and non-infectious are given by delta_LD and delta_ID respectively.” Does LD refer to latent disease? If so, the order is wrong in the above sentence.

7) Again, in Section 3.4 of the supplemental, the text goes on to talk about 80% being infectious syphilis and the remaining “10%” being non-infectious; but what about the 10% leftover? I do not understand the remaining text in this section.

8) Remaining in Section 3.4, I question the calculations, that they might require equilibrium conditions in order to apply the proportions as they do.

9) I would appreciate how these rates might relate to measurable phenomena such as a routine screening rate that everyone might experience; people with symptoms (and recent infection acquisition) would then have an additional pressure to get tested due to either symptoms or contact tracing of an infected partner.

Remaining questions:

10) How does the model deal with international travel of Swiss MSM and foreign MSM travelling to Switzerland? Can the authors weigh in on the importance of imported syphilis? Might the SHCS have data related to partner acquisition while abroad, or sex with non-Swiss while in Switzerland? ~400 syphilis diagnoses is not very many, and I wonder whether the syphilis dynamics in Switzerland might be driven more by exogeneous factors than endogenous ones.

11) The supplemental materials indicate correlation between Beta and Beta_hiv, and initSyph_h1 and initSyph_h0. This looks like a potential identifiability issue. Are the findings related to counterfactual intervention effectiveness uniform across these dimensions? More explicitly, is targeting screening to HIV+ men still recommended across this spectrum?

12) Might HIV+ MSM enrolled in the SWHD study be different from HIV+ MSM not enrolled in the study? In particular, might they have a much higher routine screening and diagnosis rate? Additionally, might their screening rate be more difficult to modify?

13) The main conclusions would be strengthened if a measure of intervention efficiency were introduced; there are different numbers of tests being administered to the HIV+/- and behavioral class groups, and increased screening among HIV+ MSM who report non-steady partners might look even better given that this is probably not a large group of people.

**Have the authors made all data and (if applicable) computational code underlying the findings in their manuscript fully available?**

Reviewer #1: None

Reviewer #2: Yes

PLOS authors have the option to publish the peer review history of their article (what does this mean?). If published, this will include your full peer review and any attached files.

Reviewer #1: No

Reviewer #2: No
---

## [Decision Letter · Decision Letter 1]

5 Oct 2021

Dear Mr Balakrishna,

We are pleased to inform you that your manuscript 'Assessing the drivers of syphilis among men who have sex with men in Switzerland reveals a key impact of screening frequency: A modelling study' has been provisionally accepted for publication in PLOS Computational Biology.

Best regards,

Benjamin Muir Althouse

Associate Editor

PLOS Computational Biology

Thomas Leitner

Deputy Editor

PLOS Computational Biology

Reviewer's Responses to Questions

**Comments to the Authors:**

Reviewer #1: All my comments have been addressed

**Have the authors made all data and (if applicable) computational code underlying the findings in their manuscript fully available?**

Reviewer #1: Yes

PLOS authors have the option to publish the peer review history of their article (what does this mean?). If published, this will include your full peer review and any attached files.

Reviewer #1: No

---

## [Editor Report · Acceptance letter]

21 Oct 2021

PCOMPBIOL-D-21-00159R1 

Assessing the drivers of syphilis among men who have sex with men in Switzerland reveals a key impact of screening frequency: A modelling study

Dear Dr Balakrishna,

I am pleased to inform you that your manuscript has been formally accepted for publication in PLOS Computational Biology. Your manuscript is now with our production department and you will be notified of the publication date in due course.

With kind regards,

Andrea Szabo
